# D1S-Neuro Program: Frequency and Risk Factors for the Development of Diabetic Neuropathy in Pediatric Patients with Type 1 Diabetes Mellitus, a Single Center Study

**DOI:** 10.3390/biomedicines13010019

**Published:** 2024-12-26

**Authors:** Marco Piccoli, Joaquin Gutierrez de Rubalcava Doblas, Patrizia Furlan, Silvia Cocchio, Alessandro Zamberlan, Gloria Panzeri, Vincenzo Baldo, Carlo Moretti

**Affiliations:** 1Pediatric Diabetes Unit, Department of Women’s and Children’s Health, University Hospital of Padua, 35128 Padua, Italy; marco.piccoli.7@studenti.unipd.it (M.P.); joaquin.gutierrezderubalcavadoblas@phd.unipd.it (J.G.d.R.D.); alessandro.zamberlan@aopd.veneto.it (A.Z.); gloria.panzeri@aopd.veneto.it (G.P.); 2Department of Cardiac, Thoracic, Vascular Sciences, and Public Health, University of Padua, 35128 Padua, Italy; patrizia.furlan@unipd.it (P.F.); silvia.cocchio@unipd.it (S.C.); vincenzo.baldo@unipd.it (V.B.)

**Keywords:** type 1 diabetes mellitus, neuropathy, pediatric

## Abstract

**Introduction:** Diabetic neuropathy is the most common long-term complication of diabetes mellitus, widely studied in the adult population, but its prevalence in children and adolescents has not yet been clearly defined. **Materials and Methods:** Diabetic patients over 11 years old and with at least 5 years of diabetes were subjected to specific tests for the screening of diabetic peripheral neuropathy (DPN) and for the diagnosis of cardiac autonomic neuropathy (CAN). Additionally, all data related to the patients’ average hemoglobin (HbA1c) levels over the last year and the past 5 years and the monitoring and insulin delivery technology used were collected. **Results:** Tests were performed on a total of 81 patients. DN diagnostic tests identified 17 patients with signs of neuropathy (21.0%), specifically 11 with DPN (13.6%) and 7 with CAN (8.6%). Data showed that the 5-year HbA1c of those diagnosed with DPN was significantly higher compared to those without a diagnosis. The analysis also highlighted that an average HbA1c level over 5 years greater than 8.5% increases the risk of DPN by 10 times. **Conclusions:** This article confirms that diabetic neuropathy begins to develop even in pediatric patients, that various nerve conduction systems may be affected, and that poorer glycometabolic control is associated with an increased risk of developing DN. These results highlight the importance of early screening and prevention through tight glycometabolic control.

## 1. Introduction

Diabetic neuropathy (DN) represents a major complication of type 1 diabetes (T1D), which can affect both the somatic and autonomic nervous system and is associated with significant morbidity and mortality [1]. While clinical complications are rarely seen among children with T1D, there is evidence that pathogenesis and early signs can develop during childhood and accelerate during puberty [2]. Diabetic peripheral neuropathy (DPN) has been well characterized in newly diagnosed adults, with prevalence rates ranging from 10 to 26% [3], but the reported prevalence of diabetic neuropathy in children and youth varies due to the use of different diagnostic tests [4]. Among youth with T1D, prevalence rates of DPN vary between 3% [5] and 57% [4], depending on the diagnostic methods used and/or criteria used to classify DPN.

Cardiac autonomic neuropathy (CAN) is the result of impaired autonomic function and subsequent nervous system imbalance of the cardiovascular system that occurs as a result of diabetes [6] and is an independent risk factor for cardiovascular mortality [7]. CAN is one of the least recognized complications of diabetes, usually detectable in asymptomatic children and adolescents [8]. In the pediatric population, CAN varies between 4% and 75% [9]. Although screening for neuropathy is recommended by the ISPAD clinical guidelines in young people with T1D from the age of 11 years with 2–5 years of diabetes duration [10], data from the US-based SEARCH study highlight that uptake of screening is suboptimal among adolescents and young adults with T1D [11].

## 2. Aim of the Study

The primary objectives of the study are to assess the frequency of diabetic neuropathy (DN) among pediatric patients affected by type 1 diabetes from at least 11 years of age and 5 years under care at the Pediatric Diabetes Unit of the University Hospital of Padua.

The secondary objective is to investigate risk factors, with a particular focus on peripheral diabetic neuropathy (DPN) and cardiac autonomic neuropathy (CAN).

## 3. Materials and Methods

An observational study was conducted on patients with diabetes under treatment at the Pediatric Diabetes Unit of the University Hospital of Padua from 2015 to 2024. Patients over 11 years of age and with at least five years of diabetes underwent tests for the screening of diabetic neuropathy (DN), specifically foot sensitivity tests for diabetic peripheral neuropathy (DPN) and tests of heart rate and blood pressure for the diagnosis of cardiac autonomic neuropathy (CAN).

### 3.1. Instruments and Tests

For DPN, all patients completed the Michigan Neuropathy Screening Instrument-Questionnaire (MNSI) [12], translated into Italian, to assess the presence of symptoms suggestive of DPN. Standardized Meteda^®^ (San Benedetto del Tronto, Italy) equipment was used to assess fine and gross tactile, vibratory, pain, and thermal sensitivity. For CAN, Meteda^®^ standardized NeurotesterAir^®^ equipment was used to measure heart rate variability: (a) Deep Breathing (DB, E:I); (b) Lying to Standing (LTS, 30:15); (c) Valsalva maneuver (V). Age-dependent cut-off values were used to define abnormal values. All data relating to the glycated hemoglobin (HbA1c) levels over the last year and the past five years, as well as the TIR (Time in Range), TBR (Time Below Range), and TAR (Time Above Range) metrics over the last 90 days, and the monitoring technology used by the patients, were collected. The technologies analyzed included multi-daily injections (MDI) and insulin pumps (Hybrid Closed Loop, HCL). For patients who changed technology in the last five years, data related to average glycated hemoglobin levels and usage duration for each technology were collected. Based on the results of the tests for the diagnosis of CAN (values of LTS, DB, V), patients were classified as “pathological”, “borderline”, and “within normal limits”. Within the pathological group, patients with DPN and those with CAN were identified. Data were anonymized before analysis.

### 3.2. Statistical Analysis

Data were presented as percentages for categorical variables and as means ± standard deviations (SDs) for continuous variables. Continuous variables were compared using Student’s *t*-test, while categorical variables were analyzed using the chi-square test. Logistic regressions were performed to evaluate which factors were significantly associated with a higher incidence of neuropathy, peripheral neuropathy, and autonomic neuropathy, adjusting for gender, age at onset, years of diabetes, average metabolic control over the last five years, and type of technology, dichotomizing between those who used the insulin pump for at least one year and those who used multi-daily injections (or the insulin pump for less than one year). Adjusted odds ratios (adjORs) with corresponding 95% confidence intervals were estimated. Through Student’s t-test, average glycated hemoglobin levels over the last year (and related TIR, TAR, and TBR metrics) were compared between patients with at least one year of pump use and those with multi-daily injections (or less than one year of pump use). Subsequently, for patients who changed technology within the last five years, average glycated hemoglobin values obtained with MDI technology alone were compared with those obtained with HCL technology alone. Before-and-after differences from the start of the new technology were compared through repeated-measures ANOVA, obtaining difference deltas and corresponding confidence intervals (Δ(95%CI)). A *p*-value ≤ 0.05 was considered statistically significant. The analyses were performed using the Statistical Package for the Social Sciences (SPSS 28.0; SPSS Inc., Chicago, IL, USA).

## 4. Results

### 4.1. Demographics and Sample Description

Eighty-one patients aged ≥11 years and with at least 5 years of diabetes were treated at the Pediatric Diabetes Unit of the University Hospital of Padua from 2015 to 15 June 2024. The mean age of the patients was 15.7 ± 2.2 years, with an average diabetes duration of 9.2 ± 3.6 years (39.5% have had diabetes for more than 10 years). Males accounted for 56.8% of the sample (Table 1).

The average glycated hemoglobin (HbA1c) for the past year and the past 5 years are 7.2 ± 0.8 and 7.3 ± 0.8, respectively, with a time in range (TIR) of 59.9% (TAR_180: 24.4 ± 6.3, TAR_250: 13.1 ± 10.9, TBR: 2.5 ± 2.3) (Table 1). At the time of the visit, 32 patients (39.5%) were using insulin pumps (31 with HCL technology and 1 with a sensor-augmented pump, SAP), and 49 were on multi-injection therapy (all using a CGM sensor, except one). Among the 32 patients with insulin pumps, only 2 had been using it for less than 1 year, while the rest had been using it for 1.1 to 5 years (average of 3 years). Specifically, 24 patients switched from MDI to HCL technology in the past 5 years (mean MDI usage: 2.9 years; mean HCL usage: 2.1 years), while 8 had only used HCL technology (Table 1 and Table 2).

### 4.2. Diabetic Neuropathy

The diagnostic tests for diabetic neuropathy (DN) identified 17 patients (21.0%) with neuropathy, specifically 11 with peripheral neuropathy (DPN) (13.6%) and 7 with autonomic neuropathy (CAN) (8.6%). Among these, 10 had abnormal peripheral sensitivity and normal autonomic function, 6 had normal peripheral function and abnormal autonomic function, and 1 patient had both peripheral and autonomic dysfunction (VT + LTS) (Table 3).

At the physical inspection, no patient showed food deformities, dry skin, calluses, infections, and fissures. Only one patient showed symptoms indicative of DPN on the MNSI questionnaire, answering yes to question numbers 1–5 and 12; however, DPN tests were within normal limits.

The autonomic function test values for the 17 neuropathic patients compared to non-neuropathic patients were LTS: 1.34 ± 0.15 vs. 1.62 ± 0.37; IB: 1.58 ± 0.25 vs. 1.60 ± 0.17; V: 1.87 ± 0.45 vs. 1.92 ± 0.25. Specifically, three patients had abnormal LTS values, one had abnormal DB, and four had abnormal V values. Regarding peripheral function, 11 had impaired foot sensitivity, specifically 7 to pain, 2 to vibration, 1 to both vibration and pain, and 1 to both vibration and monofilament (Table 3).

Three patients (3.7%) had borderline autonomic function values (mean LTS: 1.38 ± 0.17; mean DB: 1.53 ± 0.22; mean V: 1.77 ± 0.26) and normal peripheral function (Table 1). Overall, DN was diagnosed in 21.0% (n = 17) of the total diabetic patients, specifically 21.7% of males (n = 10), 28.1% (n = 9) of patients aged 17–20, and 25.0% (n = 8) of those who had diabetes for more than 10 years (Table 4).

The bivariate analysis showed that patients diagnosed with DN had significantly higher HbA1c levels over the past 5 years compared to those without DN (7.6 ± 0.9 vs. 7.2 ± 0.8; *p* = 0.041), while there were no significant differences in the metrics from the past 90 days (TIR: 56.6 ± 13.8 vs. 60.8 ± 14.7; TAR_180: 25.7 ± 6.0 vs. 24.1 ± 6.3; TAR_250: 14.8 ± 10.4 vs. 12.7 ± 11.1; TBR: 2.9 ± 2.5 vs. 2.4 ± 2.2). In terms of technology, 15.6% of patients using an insulin pump and 24.5% of those on multi-injection therapy were diagnosed with DN (Table 4); of these patients, 13.6% (n = 11) were diagnosed with DPN specifically, 15.2% of males (n = 7) and 11.4% of females (n = 4)—18.8% (n = 6) of patients aged 17–20 compared to 10.2 of patients aged 11–16, and 21.9% (n = 7) of those with more than 10 years of diabetes compared to 8.2% of those with less than 10 years (Table 4).

Patients diagnosed with DPN had significantly higher HbA1c levels over the past 5 years compared to those without DPN (7.8 ± 1.1 vs. 7.3 ± 0.7; *p* = 0.049), while there were no significant differences in the metrics from the past 90 days (TIR: 59.8 ± 11.4 vs. 59.9 ± 15.0; TAR_180: 24.7 ± 5.6 vs. 24.4 ± 6.4; TAR_250: 12.2 ± 7.3 vs. 13.3 ± 11.4; TBR: 3.3 ± 2.5 vs. 2.4 ± 2.2). DPN was diagnosed in 16.3% (n = 8) of MDI patients and 9.4% (n = 3) of insulin pump users (Table 5). CAN was found in seven patients, or 8.6% of the total, including 6.5% of males (n = 3) and 11.4% of females (n = 4), 9.4% (n = 3) of patients aged 17–20, 12.2% (n = 6) of those with less than 10 years of diabetes, and 8.2% (n = 4) of those using multi-injection therapy (Table 4). The average HbA1c for CAN patients was 7.5 ± 0.9 for the past year and 7.3 ± 0.3 for the past 5 years, with no significant differences. However, CAN was significantly associated with the age of diabetes onset; patients diagnosed with CAN had a later onset of the disease compared to others (9.9 ± 2.5 years vs. 6.3 ± 3.3 years; Δ: 3.6 95% CI (1.08; 6.14), *p* = 0.006) (Table 5).

The correlation between neuropathy and glycated hemoglobin was confirmed by multivariate analysis in Table 5. It showed that the risk of DPN is significantly higher in patients with HbA1c of 5 years > 8.5 compared to those with HbA1c of 5 years ≤ 8.5 (50% vs. 10.7%, adjOR (95% CI): 10.22 (1.19–87.68)). Specifically, 75 patients had HbA1c of 5 years ≤ 8.5, and 6 had HbA1c of 5 years > 8.5. DPN prevalence was 10.7% among the former (8/75) and 50% (3/6) among the latter. A 5-year average HbA1c > 8.5% increased the risk of DPN by 10 times.

The six patients with 5-year average HbA1c > 8.5 had diabetes for an average of 11 years (four for more than 10 years and two for 9 years). Four of them had been using insulin pumps for at least 1 year (an average of 1.8 years), one was only using multi-injection therapy, and one had been using the pump for less than 1 year.

Three developed peripheral neuropathy (pain sensitivity), but none developed autonomic neuropathy. Their average 5-year HbA1c was 9.2%, which resulted from 3.2 years of multi-injection therapy with a mean HbA1c of 10% and 1.8 years of pump therapy with an average HbA1c of 7.2. Thus, multi-injection therapy contributed most to poor metabolic control, but HbA1c improved significantly with the technology switch.

This is due to the greater commitment and the increased difficulty in managing the insulin administration method that this entails. Regarding CAN, age at onset is confirmed to be a significant risk factor; the risk of CAN increases significantly with the increase in age at diabetes onset (9.9 ± 2.5 vs. 6.3 ± 3.3, adjOR (95% CI): 1.86 (1.05–3.29)) (Table 5).

Table 6 compares glycated hemoglobin and metrics from the past year between those using multiple daily injections (MDI) therapy and those using hybrid closed-loop (HCL) technology. It highlights the significant contribution of insulin pump use in improving metabolic control, specifically in terms of lowering glycated hemoglobin levels. The average HbA1c over the past year for those using the insulin pump (for at least one year) is significantly lower compared to those using multiple daily injections (6.83 vs. 7.49; Δ = 0.66 (−0.98; −0.33)). Specifically, those who have been using the pump for at least 1 year have a significantly higher Time in Range (TIR) (70.43 vs. 53.62; Δ = 16.81 (11.27; 22.36)) and a significantly lower Time Above Range (TAR); specifically, TAR_180 (20.13 vs. 26.98; Δ = −6.85 (−9.29; −4.39)) and TAR_250 (7.13 vs. 16.74; Δ = −9.61 (−14.18; −5.03)). No differences were found in terms of the Time Below Range (TBR) (Table 6).

In the last 5 years, 24 subjects have switched from multiple daily injections to insulin pumps. Comparing their glycated hemoglobin levels before and after the technology switch, the effectiveness of HCL technology in improving metabolic control becomes evident. As shown in Table 7, overall, the average HbA1c values decreased from 7.6 to 6.8 (Δ = −0.80 (−1.52; −0.33)). In particular, it significantly decreased for females (Δ = −0.95 (−2.01; −0.27)), for those with diabetes for more than 10 years (Δ = −1.55 (−2.58; −0.52)), and for individuals aged 17–20 years (Δ = −1.77 (−2.66; −0.87)). It is confirmed that the technology switch improved HbA1c values in subjects who had an average HbA1c_5 years > 8.5 over the previous 5 years. The HbA1c levels of those diagnosed with neuropathy also decreased (Δ = −2.13 (−3.43; −0.82)), demonstrating how the progressive neural damage was caused by poor glycemic control that persisted during multiple daily injection therapy.

## 5. Discussion

Regarding diabetic neuropathy, starting in early 2024, a screening was conducted for the first time, including 81 patients followed by the Pediatric Diabetes Unit at Padua, aged over 11 years and with type 1 diabetes for at least 5 years. These patients underwent screening tests for peripheral neuropathy and diagnostic tests for cardiac autonomic neuropathy, standardized at the international level and widely used to study the adult population.

The results confirm that diabetic neuropathy begins to develop even in pediatric patients and that various nerve conduction systems can be affected. This article found that 21% of the studied population showed signs of diabetic neuropathy, with 13% showing signs of DPN and 8.6% showing signs of CAN. These data are consistent with the literature, where the prevalence of DPN is reported to range from 3% [5] to 57% [4], depending on the methodologies used. Similarly, these results for CAN align with the literature, with an estimated prevalence in the pediatric population ranging from 4% to 75% [9].

None of the studied patients had a prior diagnosis of clinical neuropathy, highlighting the importance of early systematic screening and the prevention of various types of neuropathies to prevent their progression and avoid severe consequences, such as pain, reduced quality of life, increased risk of cardiovascular events, limb amputation, and mortality.

At present, however, there is no international consensus on the best tests and diagnostic criteria for the pediatric population. The most commonly used test to confirm the presence of DPN is electromyography [13], which is an invasive, expensive, and painful procedure that only studies the conduction of large myelinated fibers, failing to detect damage to small or unmyelinated nerve fibers [14]. The best diagnostic test for detecting small fibers neuropathy is skin biopsy, but no prior study has estimated small fibers diabetic neuropathy, most likely because it is invasive [15]. Therefore, the tests performed in this study may be more suitable for detecting subclinical neuropathy, which is typical in pediatric patients.

These data show that worse glycometabolic control over the past 5 years (reported as average glycated hemoglobin) is associated with a higher risk of testing positive for diabetic neuropathy, particularly for DPN. This study found that patients with HbA1c greater than 8.5% have a 10 times higher risk of showing signs of DPN compared to those with HbA1c lower than 8.5%. This finding is supported by several high-quality studies that report a correlation between probable or confirmed DPN in patients aged at least 10.1–11.5 years and poor glycemic control [16,17,18].

As for CAN, these data did not show a statistically significant correlation between poorer glycometabolic control and the presence of CAN. This result could be due to the small sample size analyzed, as there is a non-statistically significant trend showing higher average HbA1c values over the past year in patients with signs of CAN. Additionally, these data found an association between positive test results and a later age at disease onset. This finding is not supported by the literature but could be explained by reduced neural plasticity in patients with a later onset age, resulting in irreversible damage due to hyperglycemia at disease onset.

Currently, there are no therapeutic options for diabetic neuropathy (whether DPN or CAN) in patients under 25 years of age [19], so the only tool available to the clinician is prevention through strict control of risk factors, which can halt the natural progression of the disease. Since age at disease onset and diabetes duration are non-modifiable risk factors, it is crucial to maintain strict glycometabolic control to prevent these complications or, if present, to halt their progression.

Regarding the use of advanced HCL devices for insulin administration, this study confirms, as reported in numerous studies, the better glycometabolic control in patients using these devices compared to those on multiple daily injections. This article also shows that patients who switch from multiple daily injections to insulin administration via HCL pumps experience a significant improvement in glycometabolic control, with a marked reduction in HbA1c values, including those who tested positive for DPN and CAN. This study did not find a correlation between the therapy used and the detection of diabetic neuropathy, likely because a significant number of patients started HCL therapy less than 5 years ago, thus not experiencing a lasting benefit in glycometabolic control.

However, based on this analysis, it can be deduced that long-term glycometabolic control through these devices can play a key role in preventing the development of diabetic neuropathy, thus reducing the morbidity and mortality associated with this chronic disease.

## 6. Conclusions

Results confirm that diabetic neuropathy begins to develop even in pediatric patients, that various nerve conduction systems may be affected, and that poorer glycometabolic control is associated with an increased risk of developing DN, specifically for DPN. This article demonstrated the importance of early screening of both DPN and CAN and the importance of tight glycemic control.

## 7. Limitations

The main limitation of the study is that it was designed as a single-center study, which, therefore, presents a limited sample size. Moreover, the study is a retrospective study without randomization and without a control group.

## Figures and Tables

**Table 1 biomedicines-13-00019-t001:** Description of the sample.

	Total (n.81)	Pathological (n.17)	Borderline (n.3)	Normal (n.61)
Sex				
Males	46 (56.8%)	10 (58.8%)	2 (66.7%)	34 (55.7%)
Females	35 (43.2%)	7 (41.2%)	1 (33.3%)	27 (44.3%)
Age ± SD	15.7 ± 2.2	16.5 ± 1.9	15.3 ± 1.9	15.6 ± 2.3
Years of Diabetes ± SD	9.2 ± 3.6	9.2 ± 3.8	9.3 ± 4.5	9.2 ± 3.6
Glycated Hemoglobin				
HbA1c_1year ± SD	7.2 ± 0.8	7.3 ± 0.8	7.6 ± 0.3	7.2 ± 0.8
HbA1c_5years ± SD	7.3 ± 0.8	7.6 ± 0.9	7.4 ± 0.7	7.2 ± 0.8
HbA1c_Previous5years ± SD	7.5 ± 0.7	7.8 ± 0.9	-	7.4 ± 0.6
Metrics (90 days)				
TIR ± SD	59.9 ± 14.5	56.6 ± 13.8	54 ± 10.6	61.2 ± 14.8
TAR_180 ± SD	24.4 ± 6.3	25.7 ± 6.0	27.0 ± 1.7	23.9 ± 6.4
TAR_250 ± SD	13.1 ± 10.9	14.8 ± 10.4	15.7 ± 9.1	12.5 ± 11.2
TBR ± SD	2.5 ± 2.3	2.9 ± 2.5	3.3 ± 4.2	2.4 ± 2.1
Technology at the visit				
Insulin Pump	32 (39.5%)	5 (29.4%)	1 (33.3%)	26 (42.6%)
HCL	31 (38.3%)	5 (29.4%)	1 (33.3%)	25 (41.0%)
SAP	1 (1.2%)	-	-	1 (1.6%)
Multiple daily injections (MDI)	49 (60.5%)	12 (70.6%)	2 (66.7%)	35 (57.4%)
MDI with sensor	48 (59.3%)	12 (70.6%)	2 (66.7%)	34 (55.7%)
MDI without sensor	1 (1.2%)	-	-	1 (1.6%)

**Table 2 biomedicines-13-00019-t002:** Distribution of the sample by the technology used.

**Technology at the visit**			**n.81**	**(%)**
Insulin Pump			32	(39.5%)
Multiple daily injections (MDI)			49	(60.5%)
Insulin Pump for at least 1. year			30	(37.0%)
MDI only or insulin pump for less than 1 year			51	63.0%
**Change of technology (last 5 years)**	**n**	**(%)**	**Average time (years)**	**Average HbA1c**
MDI → HCL	24	29.6%	MDI: 2.9	7.6
			HCL: 2.1	6.8
HCL only	8	9.9%	5	7.1
MDI only	5	7.4%	5	7.4

**Table 3 biomedicines-13-00019-t003:** Distribution of the sample by results of autonomic and peripheral functionality tests.

Foot Sensitivity	Autonomic Functionality						Total
	Normal	Pathological	DB	LTS	LTS + V	V	
Normal	64	6	1	1	1	3	70
Pathological	10	1					11
Painful	7	0					7
VT	1	1		1			2
VT + Painful	1	0					1
VT + Monofilament	1						1
Total	74	7	1	2	1	3	81

**Table 4 biomedicines-13-00019-t004:** Distribution of the sample by diagnosis of ND, NPD and NAD—bivariate analysis. ns means non-significant.

	Total	Neuropathy (n.17)					Peripheral Neuropathy (n.11)					Autonomic Neuropathy (n.7)				
		Yes		No		*p*	Yes		No		*p*	Yes		No		*p*
		n.	(%)	n.	(%)		n.	(%)	n.	(%)		n.	(%)	n.	(%)	
Total	81	17	(21.0)	64	(79.0)		11	(13.6)	70	(86.4)		7	(8.5)	74	(91.4)	
Sex																
Males	46	10	(21.7)	36	(78.3)	ns	7	(15.2)	39	(84.8)	ns	3	(6.5)	43	(93.5)	ns
Females	35	7	(20.0)	28	(80.0)		4	(11.4)	31	(88.6)		4	(11.4)	31	(88.6)	
Age Group (n, %)																
11–16 years	49	8	(16.3)	41	(83.7)	ns	5	(10.2)	44	(89.8)	ns	4	(8.2)	45	(91.8)	ns
17–20 years	32	9	(28.1)	23	(71.9)		6	(18.8)	26	(81.3)		3	(9.4)	29	(90.6)	
Age ± SD	15.7 ± 2.2	16.5 ± 1.9		15.6 ± 2.2		ns	16.2 ± 2.0		15.7 ± 2.2		ns	16.6 ± 1.9		15.7 ± 2.2		ns
Onset age ± SD	6.6 ± 3.4	7.3 ± 3.4		6.4 ± 3.3		ns	5.8 ± 2.8		6.7 ± 3.4		ns	9.9 ± 2.5		6.3 ± 3.3		0.006
Years of Diabetes (n, %)																
5–9 years	49	9	(18.4)	40	(81.6)	ns	4	(8.2)	45	(91.8)	ns	6	(12.2)	43	(87.8)	ns
10+ years	32	8	(25.0)	24	(75.0)		7	(21.9)	25	(78.1)		1	(3.1)	31	(96.9)	
Average diabetes years ± SD	9.2 ± 3.6	9.2 ± 3.8		9.2 ± 3.6		ns	10.4 ± 3.5		9.0 ± 3.6		ns	6.7 ± 3.3		9.4 ± 3.6		0.029
Glycated hemoglobin																
HbA1c_1year ± SD	7.2 ± 0.8	7.3 ± 0.8		7.2 ± 0.8		ns	7.1 ± 0.8		7.3 ± 0.8		ns	7.5 ± 0.9		7.2 ± 0.8		ns
HbA1c_5years ± SD	7.3 ± 0.8	7.6 ± 0.9		7.2 ± 0.8		0.041	7.8 ± 1.1		7.3 ± 0.7		0.049	7.3 ± 0.3		7.3 ± 0.9		ns
Metrics (90 days)																
TIR ± SD	59.9 ± 14.5	56.6 ± 13.8		60.8 ± 14.7		ns	59.8 ± 11.4		59.9 ± 15.0		ns	53.4 ± 16.9		60.6 ± 14.2		ns
TAR_180 ± SD	24.4 ± 6.3	25.7 ± 6.0		24.1 ± 6.3		ns	24.7 ± 5.6		24.4 ± 6.4		ns	26.7 ± 6.5		24.2 ± 6.2		ns
TAR_250 ± SD	13.1 ± 10.9	14.8 ± 10.4		12.7 ± 11.1		ns	12.2 ± 7.3		13.3 ± 11.4		ns	17.6 ± 13.9		12.7 ± 10.6		ns
TBR ± SD	2.5 ± 2.3	2.9 ± 2.5		2.4 ± 2.2		ns	3.3 ± 2.5		2.4 ± 2.2		ns	2.3 ± 2.3		2.6 ± 2.3		ns
Technology at visit																
Insulin Pump	32	5	(15.6)	27	(84.4)	ns	3	(9.4)	29	(90.6)	ns	3	(9.4)	29	(90.6)	ns
Multi.injection therapy	49	12	(24.5)	37	(75.5)		8	(16.3)	41	(83.7)		4	(8.2)	45	(91.8)	

**Table 5 biomedicines-13-00019-t005:** Distribution of subjects with neuropathy by main risk factors—logistic multivariate analysis. ns means non-significant.

	**Total (n.81)** **n.**	**Neuropathy (n.17)**		
		**n. (%)**	**adjOR (IC95%)**	** *p* **
Sex				
Males	46	10 (21.7)	0.82 (0.24–2.74)	0.741
Females	35	7 (20.0)	Ref	
Onset age	6.6 ± 3.4	7.3 ± 3.4	1.28 (0.97–1.7)	0.085
Years of diabetes				
05–09	49	9 (18.4)	Ref	
10+	32	8 (25.0)	4.41 (0.60–32.49)	0.145
Metabolic Control				
HbA1c_5years <= 8.5	75	14 (18.7)	Ref	
HbA1c_5years > 8.5	6	3 (50.0)	3.75 (0.55–25.62)	0.177
Technology				
Insulin Pump for at least 1 year	30	5 (16.7)	Ref	
MDI/Insulin Pump for less than 1 year	51	12 (23.5)	1.66 (0.45–6.17)	0.450
	**Total (n.81)** **n.**	**Peripheral Neuropathy (n.11)**		
		**n. (%)**	**adjOR (IC95%)**	** *p* **
Sex				
Males	46	7 (15.2)	1.08 (0.24–4.81)	0.919
Females	35	4 (11.4)	Ref	
Onset age	6.6 ± 3.4	5.8 ± 2.8	0.99 (0.72–1.37)	0.973
Years of diabetes				
05–09	49	4 (8.2)	Ref	
10+	32	7 (21.9)	2.17 (0.24–19.32)	0.486
Metabolic Control				
HbA1c_5years <= 8.5	75	8 (10.7)	Ref	
HbA1c_5years > 8.5	6	3 (50.0)	10.22 (1.19–87.68)	0.034
Technology				
Insulin Pump for at least 1 year	30	3 (10.0)	Ref	
MDI/Insulin Pump for less than 1 year	51	8 (15.7)	2.37 (0.41–13.56)	0.332
	**Total (n.81)** **n.**	**Autonomic Neuropathy (n.7)**		
		**n. (%)**	**adjOR (IC95%)**	** *p* **
Sex				
Males	46	3 (6.5)	Ref	
Females	35	4 (11.4)	2.67 (0.44–16.19)	0.286
Onset age	6.6 ± 3.4	9.9 ± 2.5	1.86 (1.05–3.29)	0.034
Years of diabetes	9.2 ±3.6	6.7 ± 3.3	1.22 (0.69–2.13)	0.490
Metabolic Control				
HbA1c_5years <= 8.5	75	7	Ref	ns
HbA1c_5years > 8.5	6	0	--	
Technology				
Insulin Pump for at least 1 year	30	3 (10.0)	Ref	
MDI/Insulin Pump for less than 1 year	51	4 (7.8)	0.38 (0.06–2.42)	0.303

**Table 6 biomedicines-13-00019-t006:** Differences in terms of HbA1c and metrics between those using insulin pumpsand multi-injection therapy over the past year. ns means non-significant.

	MDI/Insulin Pump for Less Than 1 Year		HCL for at Least 1 Year		Total		Δ	(IC95%)	*p*
	Mean	SD	Mean	SD	Mean	SD			
Last year Glycated Heamoglobin	7.49	0.76	6.83	0.59	7.24	0.77	−0.66	(−0.98; −0.33)	<0.001
Metrics at 90 days									
TIR	53.62	14.06	70.43	7.53	59.93	14.51	16.81	(11.27; 22.36)	<0.001
TAR	26.98	5.95	20.13	4.04	24.41	6.25	−6.85	(−9.29; −4.39)	<0.001
TAR 250	16.74	12.1	7.13	4.31	13.14	10.94	−9.61	(−14.18; −5.03)	<0.001
TBR	2.66	2.59	2.3	1.56	2.53	2.26	−0.36	(−1.40: 0.68)	ns

**Table 7 biomedicines-13-00019-t007:** Differences in HbA1c before and after the technology shift in the last 5 years (N = 24 subjects). ns means non-significant.

	HbA1c_MDI		HbA1c_HCL		Differences HbA1c_HCL-HbA1c_MDI		
	Mean	SD	Mean	SD	Δ	IC95%	*p*
Total	7.60	1.30	6.80	0.50	−0.80	(−1.52; −0.33)	0.004
Sex							
Females	7.86	1.71	6.91	0.88	−0.95	(−2.01; −0.27)	0.013
Males	7.44	0.86	6.79	0.54	−0.65	(−1.47; 0.10)	ns
Years of diabetes							
05–09	7.38	1.04	6.79	0.47	−0.59	(−1.29: 0.072)	ns
10+	8.26	1.73	6.71	0.66	−1.55	(−2.66; −0.87)	0.005
Age at the visit							
11–16	7.12	0.55	6.70	0.35	−0.43	(−1.08; 0.22)	ns
17–20	8.66	1.77	6.89	0.76	−1.77	(−2.66; −0.87)	<0.001
Metabolic control in 5 years							
HbA1c_5years <= 8.5	7.14	0.56	6.78	0.54	−0.37	(−0.70; −0.036)	0.031
HbA1c_5years > 8.5	10.10	1.14	6.71	0.48	−3.39	(−4.11; −2.66)	<0.001
Neuropathy							
No	7.39	1.13	6.77	0.46	−0.62	(−1.23; −0.037)	0.039
Yes	8.89	1.54	6.77	0.83	−2.13	(−3.43; −0.82)	0.003
Peripheral Neuropathy							
No	7.41	1.11	6.82	0.51	−0.59	(−1.13; −0.07)	0.028
Yes	9.25	1.67	6.41	0.54	−2.84	(−4.21; 1.47)	<0.001
Autonomic Neuropathy							
No	7.64	1.37	6.75	0.47	−0.89	(−1.54; −0.29)	0.006
Yes	7.65	0.23	6.97	1.20	−0.67	(−2.71; 1.36)	ns

## Data Availability

The original contributions presented in this study are included in the article. Further inquiries can be directed to the corresponding author.

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
