# Peer review of "D1S-Neuro Program: Frequency and Risk Factors for the Development of Diabetic Neuropathy in Pediatric Patients with Type 1 Diabetes Mellitus, a Single Center Study"

_biomedicines, 2024, doi:10.3390/biomedicines13010019_

Round 1
Reviewer 1 Report
Comments and Suggestions for Authors
Comments to the Authors,
I read with great interest the manuscript entitled " this manuscript entitled " D1S-Neuro Program: Prevalence and Risk Factors for the Development of Diabetic Neuropathy in Pediatric Patients with Type 1 Diabetes Mellitus ". It is an interesting topic. Attached below my comments that I hope would add to the manuscript.
Title:
" D1S-Neuro Program: Prevalence and Risk Factors for the Development of
Diabetic Neuropathy in Pediatric Patients with Type 1 Diabetes Mellitus " The term prevalence is misleading as prevalence studies should have much larger sample size with multicentric nature. Better to replace it by frequency and add single center study
Methodology:
It would be nice to add the justification of the very small sample size used.
How were the patients recruited and what were the selection criteria?
What about the associations, did any of the patients have nephropathy or retinopathy?
What about their lipids profile data?
What about the CGM metrics of those on CGM, was neuropathy related to glycemic variability, time in range, above or below range?
In which language was the Michigan Neuropathy Screening Instrument used? In English or Italian?
If you used an Italian version, was it validated?
You mentioned that Age dependent cut-off value were used to define abnormal values. Were these cut off values validated and what are their references?
Results:
The results are purely descriptive.
Table 1, it would be nice to add the P value of each comparison beside it.
Table 2: I think those on insulin pump for less than 1 year are only 2 patients.
Table 3: You are comparing those having neuropathy and not using the test used to differentiate them. It doesn't make sense.
Table 5: Interestingly no relation was found between neuropathy and glycemic control or insulin pump use. This is an odd result. How do you explain this?
Conclusion:
The use of HCL for the delivery of insulin show a better glycemic control, preventing the development of DN. This isnot justified by your results.
Limitations:
It is better to add a limitations section.
Minor comments:
It's better to avoid the terms we and our throughout the manuscript.
Comments on the Quality of English Language
Minor editing is needed.
Author Response
Dear reviewer,
Thank you for your work and for appreciating our article.
Here is the answer to the questions you raised:
1) We have changed the title of the article as you have suggested.
2) This study is a monocentric study, the small sample recruited is because of that, we have added this justification in the "limitations" section.
3) We followed the ISPAD guidelines and recruited all the patients at least 11 years old and with type 1 diabetes for at least 5 years followed in our centre.
4) All patients were tested for diabetic nephropaty and retinopathy but no one showed signs of complications.
5) We didn't have the lipid profile of all the patients, that's why we did't include it.
6) We have found no correlation between neuropathy and TIR, TAR, TBR value. We think that the development of DN takes long time (data are significant with mean HbA1c of 5 years) and we didn't have CGM data of that lenght of time.
7) It was used in italian. Sadly it's not validated, but largely used. We wrote in the article that it was administred in italian.
8) We used the validated age depend cut-off of the Meteda instruments NeurotesterAir.
9) It is just a descriptive table of the 3 groups. The differences and p values ​​are highlighted in Table 5.
10) We confirm, those on insulin pump for less than 1 year are only 2. We corrected the text in the table. Those on MDI or insulin pump for less than 1 year are 51. We corrected the mistake in that table.
11) That's right. We decided to delete table 3, that doesn't give useful informations.
12) A significant correlation was found between peripheral neuropathy and glycated hemoglobin of the last 5 years. In fact, the multivariate analysis confirmed a risk of peripheral neuropathy approximately 10 times greater in those who have a glycated hemoglobin of the last 5 years >=8.5 (Table 5 and 6). A correlation with autonomic neuropathy was not found, perhaps due to the small sample size.
Regarding technology, the insulin pump is not used by everyone for at least 5 years. Only 8 subjects use it for 5 years while the other 24 use it for an average of 2.1 years (Table 2). A longer time of use of the pump in a larger sample size would be desirable, to observe an improvement in long-term metabolic control. Furthermore, the change in technology led to a significant reduction in average glycation among subjects with HbA1c_5years >=8.5 and among those with peripheral neuropathy. Therefore it would be interesting to monitor a longer follow-up to verify the evolution of the disease.
13) You are right, we cancelled that sentence in the conclusion section.
14) We added a "limitations of the study" section.
15) We changed the draft avoiding the use of we and our.
Reviewer 2 Report
Comments and Suggestions for Authors
The authors demonstrated diabetic neuropathy can develop pediatric patients with Type 1 DM. They also showed that poor glucometabolic control can be associated with increased risk of neuropathy and tight glucometabolic control is important. Following are reviewer’s comments.
1. The authors should describe the aim of this study in the Introduction section.
2. Please cite references for the MNSI questionnaire.
3. The results of the MNSI questionnaire should be fully described.
4. Why did the authors not use the MNSI physical assessment to detect DPN?
5. The authors showed that CAN is more common in patients with a later age of onset and considered the influence of neuroplasticity. On the other hand, patients with DPN and without DPN have a similar age of onset. Does this mean that neural plasticity changes differently for somatosensory and autonomic nerves?
6. The authors should incorporate the limitations of this study in the Discussion section.
Author Response
Dear reviewer,
Thank you for your work and for appreciating our article.
Here is the answer to the questions you raised:
1) in the new draft we have added a section "Aim of the study".
2) We have added the citation for the MNSI.
3) We expanded the description of the MNSI results.
4) We used the MNSI physical assessment as a base. We have added in the article that no patient showed foot deformities, dry skin, calluses, infections and fissures. We used more precise instruments (for example a Biothesiometer and not a 128-Hz tuning fork).
5) In literature there are no evidences of differences in term of neural plasticity between somatosensory and autonomic nerves. But conditions affecting the autonomic nerves systems in pediatric patients are rare, so is still unkown how the body respond to an early damage of that system.
6) We have added a "limitations" section as requested.
Round 2
Reviewer 1 Report
Comments and Suggestions for Authors
All comments were addressed sufficiently.
Reviewer 2 Report
Comments and Suggestions for Authors
The authors replied to reviewer's comments and revised manuscript appropriately. There are no further comments.